# Bleeding Risk Factors after Endoscopic Submucosal Dissection in Early Gastric Cancer and the Necessity of “Second-Look” Endoscopic Examination on the following Day

**DOI:** 10.3390/jcm11040914

**Published:** 2022-02-09

**Authors:** Rika Kobayashi, Ken Kawaura, Tohru Ito, Sadafumi Azukisawa, Hiroaki Kunou, Junji Kamai, Kazu Hamada, Tsuyoshi Mukai, Hidekazu Kitakata, Yasuhito Ishigaki

**Affiliations:** 1Department of Gastroenterological Endoscopy, Kanazawa Medical University, Uchinada 920-0293, Ishikawa, Japan; ura-ken@kanazawa-med.ac.jp (K.K.); itotohru@kanazawa-med.ac.jp (T.I.); azuki-az@kanazawa-med.ac.jp (S.A.); kuno-h@kanazawa-med.ac.jp (H.K.); j-u-n@kanazawa-med.ac.jp (J.K.); h-kazu@kanazawa-med.ac.jp (K.H.); tsmukai@kanazawa-med.ac.jp (T.M.); kitakata@kanazawa-med.ac.jp (H.K.); 2Medical Research Institute, Kanazawa Medical University, Uchinada 920-0293, Ishikawa, Japan; ishigaki@kanazawa-med.ac.jp

**Keywords:** early gastric cancer, endoscopic submucosal dissection (ESD), second-look

## Abstract

Background and Aim: Hemorrhage is often encountered after endoscopic submucosal dissection (ESD). In addition to active bleeding after resection, exposed blood vessels and blood clots without active bleeding on the post-dissection ulcer floor have been recognized within our department. We consider exposed and/or observable vessel findings and clots on the ulcer floor after re-section as important risk factors for hemorrhage. Here, we compared and examined the active bleeding frequency and “post-resection ulcer at risk of bleeding” on the day following ESD, in relation to their risk factors. Method: We retrospectively examined 447 patients who underwent second-look endoscopy in our department between August 2008 and March 2018. Logistic regression analyses were performed to determine the hazard ratio and 95% confidence interval. We compared the association of each factor mentioned above with active bleeding on the day after ESD and the presence of ulcers at risk of bleeding after resection. Results: Our retrospective analysis revealed that the risk factors were larger ulcer sizes and the administration of antithrombotic drugs. Additionally, the risk was low for upper body lesions but high for antral lesions. Conclusion: Our results may help determine whether second-look endoscopy should be performed to minimize active bleeding after ESD, reduce postoperative complications, and improve medical safety.

## 1. Introduction

Endoscopic submucosal dissection (ESD) is the standard treatment for mucosal cancer in Japan [1,2,3,4]. Second-look upper gastrointestinal endoscopy is usually performed on the day after ESD, regardless of the presence or absence of clinical symptoms. It has been commonly used to check for bleeding from resected sites, to observe ulcer bases, and to determine when to resume eating after surgery. However, ESD can result in iatrogenic complications, even when performed by skilled endoscopists. Post-ESD hemorrhage is often encountered, and endoscopic hemostasis may be required [5,6,7,8,9,10]. However, a number of institutions do not perform upper gastrointestinal endoscopy on the day following ESD (hereinafter referred to as second-look endoscopy), and some authors have reported that second-look endoscopy is unnecessary [11,12].

In Japan, the Gastric Cancer Treatment Guidelines 2018 (published by the Japanese Gastric Cancer Association) defines the absolute indications for ESD as either macroscopic intramucosal cancer with a diameter of >2 cm (cT1a), which is a differentiated carcinoma without ulcerative findings (UL0), or macroscopic intramucosal cancer with a diameter of ≤3 cm (cT1a), which is a differentiated carcinoma with ulcerative findings (UL1). The guidelines also define the expanded indication of ESD as macroscopic intramucosal cancer with a diameter of ≤2 cm (cT1a), which is an undifferentiated carcinoma without ulcerative findings (UL0). For lesions that do not fulfill these absolute or expanded indications, and which are therefore classified as out-of-indication, gastric resection is suggested as the standard of care. However, in some cases in which gastric resection is difficult for reasons, such as older age or the presence of a comorbidity, ESD may be selected for out-of-indication lesions, considering possible lymph node involvement. In these instances, the lesions are referred to as “relative indications” for ESD.

Our department is based in a university hospital and often carries out ESD for relative indication lesions. ESD is also performed in patients with a history of antithrombotic drug administration. Second-look endoscopy is performed in all patients [5,6,13,14,15,16]. Some patients had active bleeding at the time of the second examination. Additionally, we have encountered many cases without hemorrhage but with visible vessels and blood clots on the ulcer floor after resection [17]. We considered that exposed and/or observable vessel findings and clots on the ulcer floor after resection are important risk factors for hemorrhage. Here, we retrospectively analyzed the risk factors for active bleeding on the day after ESD and the presence of ulcers at risk of bleeding after resection, with the goal of clarifying the type of cases and risk factors that might require future second-look endoscopy.

## 2. Materials and Methods

### 2.1. Definition of Second-Look (2nd-Look) and Active Bleeding

In this study, second-look endoscopy refers to upper gastrointestinal endoscopy performed the day after ESD. We retrospectively examined 447 patients who underwent second-look endoscopy in our department between August 2008 and March 2018. Active bleeding was assessed in all patients. All patients were assessed for post-resection ulcers that were at risk of active bleeding. Fifty-one patients with active bleeding during second-look endoscopy were excluded, and the remaining 396 patients were assessed for the presence of ulcers that were at risk of bleeding after resection. Patient backgrounds (Table 1) and pathological findings were determined from electronic medical records, and images and endoscopic findings were analyzed using a device that stored images of endoscopic findings (Solemio ENDO; Olympus, Tokyo, Japan).

Active bleeding is defined as an endoscopic finding of ongoing bleeding from the ulcer floor after resection or pooling of fresh blood in the stomach during second-look endoscopy. For the purpose of this study, the term “active bleeding” is interchangeable with “hemorrhage” (Figure 1). Ulcers at risk of bleeding after resection were defined as those with exposed blood vessels, adhesion of blood clots to the post-dissection ulcer floor caused by exposed blood vessels, and/or the presence of blood components on the post-dissection ulcer floor (Figure 2).

### 2.2. Risk Factors for Post-ESD Ulcer with Bleeding Risk 

The following parameters were statistically compared as risk factors for active bleeding and for the presence of ulcers at risk of bleeding after resection: age (≤74 vs. ≥75 years, based on the definition of advanced age in Japan); use of general or intravenous anesthesia; time required for resection; specimen length (major axis); lesion site (upper body, middle body, lower body, antrum, anterior wall, greater curvature, posterior wall, or lesser curvature); lesion depth (mucosal vs. submucosal invasion); histological type (differentiated vs. undifferentiated); presence or absence of one or more ulcer scars in pathological tissue specimens; excision of single or multiple lesions; and history of antithrombotic drug administration. The number of cases would have been too small for statistical analysis if antithrombotic drugs were divided into anticoagulant and antiplatelet drugs, with further subgrouping according to the number of doses (one, two, or three). Therefore, we analyzed all the antithrombotic agents. Resected specimens were required to be at least 30 mm in diameter, because we marked and excised tumors with a safety zone of approximately 5 mm beyond the 20-mm margin, which is indicated for ESD. The size of each ESD specimen was defined as the largest diameter of the flattened specimen. Since ESD specimens usually have a near-oval shape, the long axis of the oval was considered to indicate the diameter. Resected tumors with a diameter of >30 mm, including an approximate 5-mm margin, were considered to be large lesions, while those with a diameter of ≤30 mm, including the 5-mm margin, were classified as ordinary-sized lesions.

### 2.3. Statistical Analysis

Logistic regression analyses were performed to determine the hazard ratio (HR) and 95% confidence interval (95% CI). JMP 9 (SAS Institute Japan Ltd., Tokyo, Japan) was used for statistical analyses. We compared the association of each factor mentioned above with both active bleeding on the day after ESD and the presence of ulcers at risk of bleeding after resection. First, we identified all the risk factors that achieved a significance level of *p* < 0.05 in the univariate analysis and then performed a multivariate analysis to determine the independent risk factors. After 51 patients with active bleeding were excluded from the group of 447 patients who underwent second-look endoscopy, 396 were included in the analysis of post-resection ulcers at risk of bleeding. Similarly, in patients with active bleeding, a univariate analysis was performed to identify all risk factors that achieved a significance level of *p* < 0.05, followed by a multivariate analysis to determine the independent risk factors.

### 2.4. Treatment Planning in Our Department

Antithrombotic drugs such as low-dose aspirin, thienopyridine, and other antiplatelet drugs were discontinued 7, 10, and 1–3 days before ESD, respectively. Warfarin was stopped 7 days before ESD. If necessary, heparin bridging was performed and was discontinued 6 h before ESD. As a general rule, antithrombotic therapies were resumed after completion of third-look endoscopy, which was performed 1 week after resection in all patients (Figure 3).

### 2.5. ESD Method in Our Department

We first confirmed and marked the lesion area using argon plasma coagulation (APC) with a 5-mm margin; then, a 1:1 mixture of physiological saline and hyaluronic acid was injected into the area surrounding the resection region. A circumferential incision was made using a combination of a hook knife (Olympus, Tokyo, Japan) and an IT knife 2 (Olympus). After all circumferential incisions were complete, physiological saline was injected into the submucosal layer, which was then detached using the IT knife 2. Hemostasis was achieved using hemostatic forceps (Olympus). A high-frequency generator (VIO300D; VIO3; Erbe, Tokyo, Japan) was used; its mode of operation is listed in Table 2. After confirming the resection of the lesion by ESD, the vessel was cauterized with hemostats for the guttural vessel and the exudative bleeding site. After confirming that the bleeding had stopped, the procedure was completed. 

The next day, during second-look endoscopy, APC and clipping were mainly used for hemostasis [18]. A proton pump inhibitor (PPI) was infused twice daily on the day of the surgery and on the day after the surgery [19,20,21,22,23,24,25]. If there was no bleeding, patients resumed eating 2 days after resection, and the route of PPI administration was switched from intravenous to oral. Since third-look endoscopy was performed 1 week after resection, the hospitalization period was 9 days (Figure 3).

## 3. Results

### 3.1. Univariate Analysis

On second-look endoscopy, 51 (11.4%) of the 447 patients demonstrated active bleeding (Table 1). In 50 of these patients, endoscopic hemostasis was achieved with APC, hemostat, and/or clipping. Open surgery was required in only one patient. The univariate analysis showed that the significant risk factors for active bleeding were upper gastric lesions, larger specimen sizes, and a history of antithrombotic drug administration (Table 1 and Table 3). Incidentally, in Japan, post-ESD bleeding is generally reported to be approximately 5%, and the bleeding rate is high at our institution.

### 3.2. Multivariate Analysis

The significance level of *p* < 0.05 demonstrated a statistically significant lower risk. Of the 51 patients who underwent second-look endoscopy and had active bleeding, 39 (76.5%) (HR = 2.736, *p* = 0.0019) had a resected specimen length >30 mm, indicating that a larger size was positively correlated with a bleeding risk. There was a 1.047 times increase in the HR with every 1-mm increase in the resected specimen size (95% CI, 1.026–0.068, *p* < 0.0001). Sixteen of the 51 (31.4%) patients with a history of antithrombotic drug administration demonstrated active bleeding despite discontinuation of the drug(s), with an HR of 1.957 (*p* = 0.0477). The multivariate analysis of the risk factors for active bleeding showed HRs of 0.265 (*p* = 0.0317), 2.582 (*p* = 0.0054), and 1.975 (*p* = 0.0487) for upper body lesions, resected specimen lengths >30 mm, and a history of antithrombotic drug administration, respectively, each of which was found to be an independent risk factor (Table 4).

In clinical practice, it is important to identify the risk factors for post-ESD active bleeding or “ulcers with a risk of bleeding”; therefore, patients with either condition were analyzed together. The univariate analysis showed that upper body lesions, antral lesions, specimen length, and a history of antithrombotic drug administration were significant factors associated with the risk of bleeding after dissection (Table 5). Only seven patients (5.0%) had upper body lesions that were susceptible to bleeding (HR = 0.280; *p* = 0.0006), indicating a significantly lower risk. Conversely, 62 (13.9%) of the 139 patients were found to have antral lesions (HR = 1.805; *p* = 0.0051) that were susceptible to bleeding, indicating a significantly higher risk. Ninety-eight (70.5%) of the 139 patients (HR = 2.329; *p* < 0.0001) with a specimen length > 30 mm had ulcers at risk of bleeding. Thirty-seven (8.3%) of the 139 patients with a history of antithrombotic drug administration demonstrated active bleeding despite discontinuation of the drug(s), with an HR of 1.706 (*p* = 0.0299). The multivariate analysis showed HRs of 0.348 (*p* = 0.0084), 1.622 (*p* = 0.0229), 2.307 (*p* = 0.0001), and 1.792 (*p* = 0.0230) for upper body lesions, antral lesions, resected specimen lengths >30 mm, and a history of antithrombotic drug administration, respectively; each of these was found to be an independent risk factor (Table 6). Independent of the combination, similar results are shown in Table 3, Table 4, Table 5 and Table 6.

## 4. Discussion

### 4.1. The Need for Second-Look after ESD 

With the development of various devices such as the hook knife, IT knife 2, and dual knife, and improvements in high-frequency electrosurgical units, together with enhanced resection techniques, mucosal cancer can be reliably and rapidly removed by ESD. In our department, we perform second-look endoscopy in all cases to screen for bleeding, and the findings are used to determine when to resume oral food consumption. In addition, during second-look procedures, we focus on exposed blood vessels and blood clots that adhere to the ulcer floor, as these are potential risk factors for bleeding. Some institutions do not perform second-look endoscopy, which is understandable, because it places a heavy burden on both patients and medical staff [11,12]. In this study, we investigated active bleeding after ESD and ulcers that were at risk of bleeding to determine the necessity of second-look endoscopy.

### 4.2. Percentage of Bleeding Complications in Our Department 

In our department, the incidence of bleeding on the day after ESD was 11.4% (of the total 497 patients, active bleeding in 51), which was higher than previously reported. In our experience, postoperative hemorrhage occurs despite the careful use of hemostatic forceps at the end of ESD. The present study showed that the risk of bleeding was lower for upper body lesions than for lesions in other sites and that a longer resected specimen length was associated with a greater risk of bleeding. In addition, the risk of postoperative hemorrhage was increased in patients with a history of antithrombotic drug administration (Table 4 and Table 6), despite the fact that such agents were discontinued prior to ESD in all cases, in accordance with the relevant guidelines. This is presumably due to the characteristics of our university hospital, where there are fewer cases of lesions with absolute indications for early-stage gastric cancer ESD than at other institutions, and many cases are handled with expanded indications.

### 4.3. Upper Body Lesions Are Low-Risk Sites

The risk of bleeding from upper body lesions was small (Table 4 and Table 6). This may have been due to the effects of stomach acid and the properties of blood vessels in the gastric mucosal surface layer in this area. Since gastric acid can damage the submucosa or muscle layer exposed after resection, leading to further ulcers, PPIs are administered perioperatively. However, these drugs do not completely block gastric acid production. Secreted gastric acid flows from the body of the stomach into the gastric antrum. While the concentration of acid in both sites is thought to be the same, it persists for a longer duration in the gastric antrum. Therefore, it is assumed that the risk of postoperative hemorrhage is smaller for ulcers in the upper body, as they are exposed to gastric acid for a shorter period than those in the antrum. Although ESD of upper body lesions frequently causes bleeding, in most cases, this bleeding arises from thin, submucosal blood vessels rather than thick vessels. Since bleeding from thin blood vessels can be stopped easily during and after resection, the risk of bleeding in the upper body is low.

### 4.4. The Larger the Ulcer, the Higher the Risk of Bleeding 

At the beginning of this study, we planned to define the resected sample size based on the area of the ulcer after resection. However, the specimens were distorted, and it was not possible to accurately calculate their areas. Therefore, we used the length of the major axis of each specimen, which can be easily measured. The target resection margin was set at 30 mm to establish a safety zone of 5 mm surrounding a 20-mm margin. A longer major axis was associated with a larger resected area and a higher risk of bleeding after resection. Previous results have indicated that a larger resection area was correlated with an elevated risk of bleeding following ESD [26].

In the current study, stomach acid caused greater damage to the ulcer floor in cases of patients with a larger resected area (Table 4 and Table 6). Post-resection hemostasis was likely incomplete for larger ulcers. While ESD resection of intramucosal carcinoma is possible regardless of tumor size, the results of the present study suggest that the area of mucosal involvement should be determined carefully and accurately, and the extent of resection should be limited to the minimum necessary.

### 4.5. Antithrombotic Drug Administration can Be a Risk Factor 

Antithrombotic drug administration can be a risk factor for postoperative hemorrhage [27,28]. Some reports have shown that the frequency of bleeding after ESD is increased even when antithrombotic drugs are discontinued [28]. Patients receiving anticoagulants should be followed up carefully, and the timing of antithrombotic drug resumption should be carefully considered.

The length of the specimen’s major axis and a history of antithrombotic drug administration were independent risk factors for active bleeding on the day following ESD. Although the multivariate analysis showed that upper body lesions were associated with a lower risk of bleeding, careful follow-up is necessary in patients who have undergone extensive resection and those with a history of antithrombotic drug administration.

In addition to active bleeding on the day after ESD, we also considered that exposed blood vessels at the base of the ulcer after resection, blood vessels with adherent clots, and blood components adhering to the ulcer base were important findings, since there is a risk of bleeding several hours after ESD or even on the following day. We performed hemostatic preventive measures in patients with ulcers at risk of bleeding after resection. As a specific prophylactic measure, even vessels without bleeding were prophylactically cauterized with APC and hemostats, and efforts were made to ensure that there were no presumed vessels at the bottom of the ESD ulcer.

### 4.6. High Risk in Antral Lesions 

In this study, the incidence of ulcers at risk of bleeding was 22.2% (of the 396 patients, there were post-resection ulcers at risk of bleeding in 88). Among the post-resection ulcers at risk of bleeding, 45.5% had antral lesions (of the 88 patients, there were antral lesions in 40). It was difficult to definitively determine whether bleeding would occur later. The risk was low for upper body lesions but high for antral lesions.

### 4.7. Risk Factors for Active Bleeding and “Post-Resection Ulcer at Risk of Bleeding”

The risk of bleeding was low for upper body lesions but high for antral lesions. A large ulcer floor ≥ 30 mm after resection was also a risk factor for bleeding. Antithrombotic drug administration can be a risk factor for postoperative hemorrhage. In the multivariate analysis, antral lesions, specimen length >30 mm, and antithrombotic drug administration were independent risk factors for ulcers at risk of bleeding after resection, as well as for active bleeding. Careful observation during and after ESD should focus on these factors. 

In this study, we found that the incidence of active bleeding after ESD and that of ulcers at risk of bleeding after resection were both higher than expected. Of the 447 patients who underwent ESD, 139 (31.1%) had one of these conditions. Four of the 447 patients (0.89%) experienced hemorrhage after hospital discharge. This study was able to identify risk factors for active bleeding and ulcers at risk of bleeding after resection; in the future, only patients with these risk factors should undergo second-look endoscopy after ESD. This may reduce the burden on both patients and healthcare professionals. We continue to believe that second-look endoscopy is effective for preventive hemostasis and is required to safely discharge at-risk patients. Authors should discuss the results and how they can be interpreted from the perspective of previous studies. These findings and their implications should be discussed in the broadest possible context. Future research directions are also highlighted.

## 5. Conclusions

Given the trend toward shorter hospital stays and higher rates of antithrombotic drug administration, we consider that second-look endoscopy following ESD is necessary for the safe and secure discharge of certain patients. Our retrospective analysis revealed that larger ulcer sizes and antithrombotic drug administration were risk factors for bleeding after ESD. In addition, the risk was low for upper body lesions but high for antral lesions. Our results may help reduce postoperative problems, improve medical safety, and aid physicians in deciding whether second-look endoscopy should be performed to minimize active bleeding after ESD. In the future, we hope to conduct randomized controlled trials of bleeding in patients with post-ESD ulcer bases who also have risk factors and that the results of these analyses will conclude that only patients with risk factors should undergo second-look endoscopy after ESD.

## Figures and Tables

**Figure 1 jcm-11-00914-f001:**
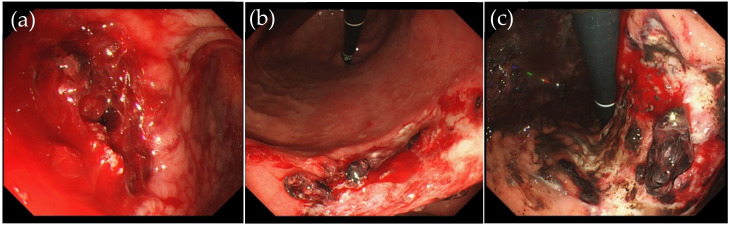
Cases with active bleeding after endoscopic submucosal dissection. Active bleeding (**a**). Oozing bleed (**b**,**c**).

**Figure 2 jcm-11-00914-f002:**
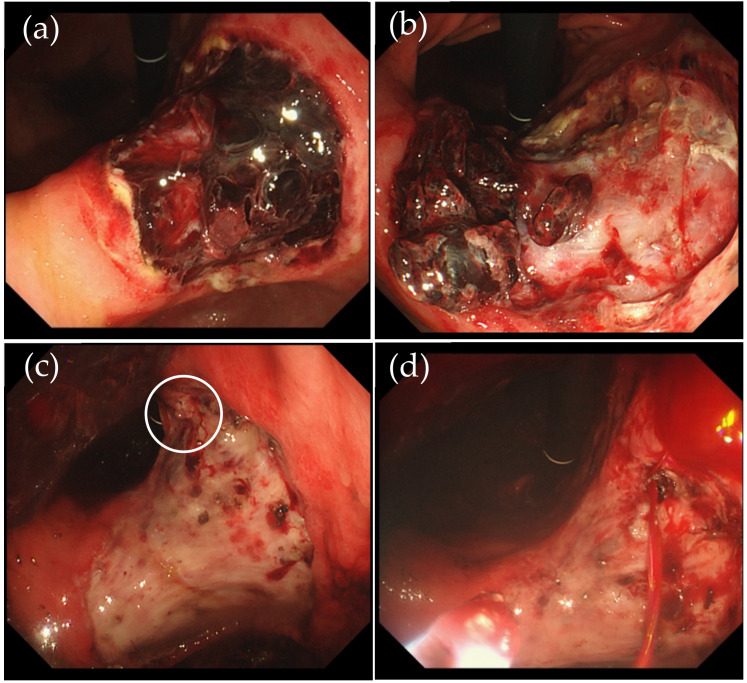
Cases of “post-resection ulcer at risk of bleeding”. Adhesion of blood clots to the post-dissection ulcer floor (**a**,**b**). Exposed blood vessels, and/or the presence of blood components on the post-dissection ulcer floor (**c**,**d**). White circle indicates a visible vessel.

**Figure 3 jcm-11-00914-f003:**
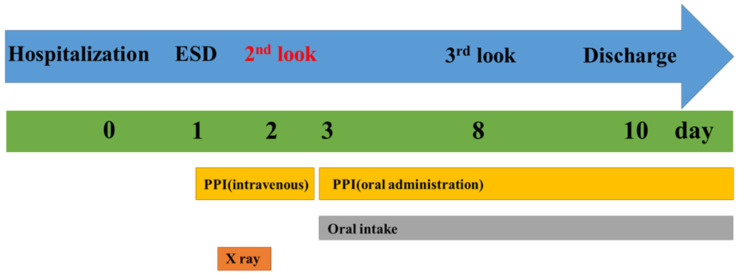
Medical treatment plan in our division (clinical path).

**Table 1 jcm-11-00914-t001:** Characteristics of patients, frequency of active bleeding, and frequency of “ulcer with a risk of bleeding”.

		Active Bleeding 51 Cases			Ulcer with a Risk of Bleeding 88 Cases		
Variable		*n* = 447	Absent	Present	*n* = 396	Absent	Present
Age							
	≤74	256	231	25	231	179	52
	≥75	191	165	26	165	129	36
	Average 72.6, SD 8.76						
Anesthesia method							
	Intravenous anesthesia	261	230	31	230	183	47
	General anesthesia	186	166	20	166	125	41
Location							
	Upper	56	54	2	54	49	5
	Middle	55	50	5	50	40	10
	Lower	179	157	22	157	124	33
	Antrum	157	135	22	135	95	40
Location							
	Anterior wall	69	61	8	61	46	15
	Greater Curvature	86	76	10	76	60	16
	Posterior wall	97	89	8	89	73	16
	Lesser Curvature	195	170	25	170	129	41
Endoscopic findings							
	Elevated type	182	162	20	162	124	38
	Depressed type	265	234	31	234	184	50
Resecting time							
	≤60 min	224	201	23	201	152	49
	>60 min	223	195	28	195	156	39
Major axis of specimen							
	≤30 mm	193	181	12	181	152	29
	>30 mm	254	215	39	215	156	59
Depth							
	Mucosal tumor	390	347	43	347	266	81
	Submucosal tumor	57	49	8	49	42	7
Ulcer scars in pathological tissue specimens							
	Negative	406	360	46	360	279	81
	Positive	41	36	5	36	29	7
Histology							
	Differentiated type	424	377	47	377	292	85
	Undifferentiated type	23	19	4	19	16	3
Antithrombotic drugs use ^‡^							
	No	356	321	35	321	254	67
	Yes	91	75	16	75	54	21
Number of resections							
	Singular	410	367	43	367	286	81
	Simultaneous multiple	37	29	8	29	22	7

^‡^ History of the use of antithrombotic drugs.

**Table 2 jcm-11-00914-t002:** Setup of the high-frequency electrosurgical unit (VIO3).

	Mode	Effect
Marking	Precise APC	10.0
Circumferential incision	Endo Cut Q	Effect 3 Duration 3 Interval 3
Dissection	Precise SECT	7.0
Hemostasis	Soft COAG	6.5

**Table 3 jcm-11-00914-t003:** Univariate analysis of the risk factors for active bleeding.

Study Population *n* = 447		Univariate Analysis		
Variable		χ^2^	HR ^†^ (95% CI ^§^)	*p*-Value
Age	≤74 ≥75	1.586	Referent 1.456 (0.810–2.622)	0.2080
Anesthesia method	Intravenous anesthesia General anesthesia	0.137	Referent 1.119 (0.620–2.058)	0.7177
Location	Upper Middle Lower Antrum	4.984 0.353 0.228 1.582	0.259 (0.042–0.868) 0.752 (0.252–1.822) 1.155 (0.635–2.076) 1.467 (0.804–2.643)	0.0256 0.5527 0.6332 0.2085
Location	Anterior wall Greater Curvature Posterior wall Lesser Curvature	0.003 0.005 1.310 0.677	1.022 (0.428–2.174) 1.027 (0.468–2.067) 0.642 (0.271–1.346) 1.278 (0.710–2.297)	0.9582 0.9436 0.2524 0.4106
Endoscopic findings	Elevated type Depressed type	0.054	Referent 1.073 (0.595–1.975)	0.8165
Resection time (min)	≤60 >60	0.580	Referent 1.255 (0.700–2.271)	0.4464
Major axis of specimen (mm)	≤30 >30	9.623	Referent 2.736 (1.431–5.601)	0.0019
Depth	Mucosal tumor Submucosal tumor	0.423	Referent 1.318 (0.548–2.834)	0.5155
Ulcer scars in pathological tissue specimens	Negative Positive	0.027	Referent 1.087 (0.360–2.684)	0.8693
Histology	Differentiated Undifferentiated	0.760	Referent 1.689 (0.475–4.726)	0.3833
Antithrombotic drugs use ^‡^	No Yes	3.920	Referent 1.957 (1.007–3.700)	0.0477
Number of resections	Singular Simultaneous multiple	3.477	Referent 2.354 (0.954–5.268)	0.0622

^†^ Hazard Ratio. ^§^ Confidence Interval. ^‡^ History of the use of antithrombotic drugs.

**Table 4 jcm-11-00914-t004:** Multivariate analysis of the risk factors for active bleeding.

		**Multivariate Analysis**		
**Variable**		**χ^2^**	**HR** **^†^ (95% CI** **^§^)**	***p*-Value**
Location	Middle, Lower, Antrum Upper	4.616	Referent 0.265 (0.042–0.905)	0.0317
Major axis of specimen (mm)	≤30 >30	8.335	Referent 2.582 (1.343–5.310)	0.0054
Antithrombotic drugs use ^‡^	No Yes	3.885	Referent 1.975 (1.004–3.757)	0.0487

^†^ Hazard Ratio. ^§^ Confidence Interval. ^‡^ History of the use of antithrombotic drugs.

**Table 5 jcm-11-00914-t005:** Univariate analysis of the risk factors for active bleeding and “post-resection ulcer at risk of bleeding”.

Study Population *n* = 447		Univariate Analysis		
Variable		χ^2^	HR ^†^ (95% CI ^§^)	*p*-Value
Age	≤74 ≥75	0.289	Referent 1.117 (0.745–1.672)	0.5907
Anesthesia method	Intravenous anesthesia General anesthesia	0.428	Referent 1.145 (0.763–1.715)	0.5128
Location	Upper Middle Lower Antrum	11.931 0.437 0.019 7.828	0.280 (0.113–0.598) 0.810 (0.420–1.493) 0.972 (0.643–1.460) 1.805 (1.194–2.729)	0.0006 0.5086 0.8901 0.0051
Location	Anterior wall Greater Curvature Posterior wall Lesser Curvature	0.189 0.037 0.671 1.217	1.129 (0.645–1.932) 0.951 (0.563–1.571) 0.672 (0.396–1.108) 1.254 (0.838–1.877)	0.6641 0.8470 0.1207 0.2699
Endoscopic findings	Elevated type Depressed type	0.085	Referent 0.941 (0.627–1.417)	0.7703
Resection time (min)	≤60 >60	0.230	Referent 0.907 (0.607–1.354)	0.6318
Major diameter of specimen (mm)	≤30 >30	15.779	Referent 2.329 (1.528–3.598)	<0.0001
Depth	Mucosal tumor Submucosal tumor	0.715	Referent 0.766 (0.398–1.405)	0.3976
Ulcer scars in pathological tissue specimens	Negative Positive	0.071	Referent 0.909 (0.434–1.798)	0.7897
Histology	Differentiated Undifferentiated	0.005	Referent 0.968 (0.365–2.325)	0.9438
Antithrombotic drugs use ^‡^	No Yes	4.714	Referent 1.706 (1.054–2.744)	0.0299
Number of resections	Singular Simultaneous multiple	1.610	Referent 1.573 (0.776–3.111)	0.2045

^†^ Hazard Ratio. ^§^ Confidence Interval. ^‡^ History of the use of antithrombotic drugs.

**Table 6 jcm-11-00914-t006:** Multivariate analysis of the risk factors for active bleeding and “post-resection ulcer at risk of bleeding”.

		Multivariate Analysis		
Variable		χ^2^	HR ^‡^ (95% CI ^§^)	*p*-Value
Location	Middle, Lower, Antrum Upper	6.952	Referent 0.348 (0.137–0.775)	0.0084
Location	Upper, Middle, Lower, Antrum	5.177	Referent 1.662 (1.073–2.581)	0.0229
Major diameter of specimen (mm)	≤30 >30	14.609	Referent 2.307 (1.496–3.605)	0.0001
Antithrombotic drugs use ^‡^	No Yes	5.171	Referent 1.792 (1.085–2.948)	0.0230

^§^ Confidence Interval. ^‡^ History of the use of antithrombotic drugs.

## Data Availability

All data for this study are stored in the Department of Gastroenterological Endoscopy, Kanazawa Medical University.

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
