# Peer review of "Bleeding Risk Factors after Endoscopic Submucosal Dissection in Early Gastric Cancer and the Necessity of “Second-Look” Endoscopic Examination on the following Day"

_jcm, 2022, doi:10.3390/jcm11040914_

Round 1

Reviewer 1 Report

Kobayashi et al. conducted a retrospective study on bleeding risk factors after endoscopic submucosal dissection in early gastric cancer and the necessity of second-look endoscopic examination on the following day. The collected data appears to be of good quality. The authors identify risk factors for active bleeding after resection and conclude that only patients with these risk factors should undergo second-line endoscopy after ESD. It seems to be the golden key to unlocking the problem. The authors should be commended for the detail of this work. This type of fieldwork should not be underestimated. Overall: a good work, well written and thought out, relates to other retrospective studies published this year, which means that the topic is up-to-date and needed. It's an excellent article and I enjoyed reading every word of it. Congratulations. 
A little note. The surname of the first author should be capitalized.

Author Response

Dear Editor of Journal of Clinical Medicine,

We thank you greatly for your reply and for reviewing our manuscript submitted on December 6th, 2021 (Manuscript ID: jcm-1519452). We appreciate the constructive comments provided by the reviewers that have allowed us to substantially improve the manuscript. We are submitting a revised version. 

We appreciate your kind comments on our manuscript, which have been very encouraging to us. We modified the author name. Thank you.

Reviewer 2 Report

  1. the abstract does not have the necessary structure required for an original study
  2. The manuscript has many typographical, capitalization and grammatical errors.
  3. Introduction:

- “this procedure can result in iatro- genic complications, even when performed by skilled endoscopists”: kindly explain what you mean by “this procedure”. ESD or second-look endoscopy?

- The research question is not straight-forward.

  1. Materials and methods

- “Fifty-one patients with active bleeding were excluded”: the aim of this study was to identify cases in which a second-look endoscopy should be performed if there are no obvious clinical signs of bleeding. Therefore, cases detected only by endoscopy without obvious clinical manifestations should not be excluded.

- Table 1 is not appropriate and should follow the objectives of the study as stated above.

-The definitions of “active bleeding” and “ulcer with a risk of bleeding” need to be defined.

- Why choose to divide the age threshold as 75?

- Table 1: what is “UL”?

5- The way the authors analyzed was not practically useful. In clinical practice, it is important to identify risk factors of after-ESD GI active bleeding or “ulcer with a risk of bleeding”. It is because endoscopic intervention is required in both cases. Therefore, patients with active bleeding and “ulcer with a risk of bleeding” should be combined and analyzed.

6- Conclusion: the conclusion did not answer the research question specifically.

Author Response

Dear Editor of Journal of Clinical Medicine,

We thank you greatly for your reply and for reviewing our manuscript submitted on December 6th, 2021 (Manuscript ID: jcm-1519452). We appreciate the constructive comments provided by the reviewers that have allowed us to substantially improve the manuscript. We are submitting a revised version. 

1.the abstract does not have the necessary structure required for an original study

Response: Based on your comment, we added the following sentences: “Our retrospective analysis revealed that risk factors were larger ulcer size and administration of antithrombotic drugs . In addition, the risk was low for upper body lesions, but high for antral lesions. Our results may help determine whether to perform a second-look to minimize active bleeding, reduce postoperative problems, and improve medical safety.”

2.The manuscript has many typographical, capitalization and grammatical errors.

Response: Thank you for helpful comment. The English text of our manuscript has again been edited by Zenis Co., Ltd. for a fee.

3.Introduction:

- “this procedure can result in iatrogenic complications, even when performed by skilled endoscopists”: kindly explain what you mean by “this procedure”. ESD or second-look endoscopy?

Response: Thank you for pointing this out. We have changed the phrase “this procedure” to “ESD.”

- The research question is not straight-forward.

Response: We changed the last sentences in the Introduction section, from “We conducted a statistical analysis to clarify the types of cases and risk factors that would require a second-look in the future, and to selectively perform a second-look in order to minimize postoperative bleeding, reduce postoperative problems, and improve medical safety. ” to “Here, we retrospectively analyzed risk factors for active bleeding on the day after ESD and the presence of ulcers at risk of bleeding after resection, with the goal of clarifying the types of cases and risk factors that might require future second-look endoscopy. ”

4.Materials and methods

- “Fifty-one patients with active bleeding were excluded”: the aim of this study was to identify cases in which a second-look endoscopy should be performed if there are no obvious clinical signs of bleeding. Therefore, cases detected only by endoscopy without obvious clinical manifestations should not be excluded.

Response: Active bleeding was examined in all 447 cases. In addition, bleeding ulcers were hidden by blood components, so the condition of the ulcer base could not always be confirmed. Therefore, the analysis of post-resection ulcers at risk of bleeding was conducted in 396 cases, after exclusion of the 51/447 cases of active bleeding.

- Table 1 is not appropriate and should follow the objectives of the study as stated above.

Response: The tables were based on the aforementioned study objectives. Some of the numbers listed were incorrect and have been corrected. The presence or absence of anesthesia was also omitted and has been added.

-The definitions of “active bleeding” and “ulcer with a risk of bleeding” need to be defined.

Response: Thank you for your pertinent comment. These definitions have been clearly presented in the Introduction section, with added text highlighted in red.

- Why choose to divide the age threshold as 75?

Response: The definition of late-stage advanced age used in Japan is 75 years old or older, which is why this cutoff was used. We added this explanation on lines 85–86 in pink highlighted text.

- Table 1: what is “UL”?

Response: UL indicates ulcer scars in pathological tissue specimens. On lines 90-91, we changed the references to the presence or absence of one or more ulcers to the presence or absence of one or more ulcer scars in pathological tissue specimens. This change is marked by pink highlighted text. The notation of UL in Tables 1, 3, 5, and 7 has also been changed.

5.The way the authors analyzed was not practically useful. In clinical practice, it is important to identify risk factors of after-ESD GI active bleeding or “ulcer with a risk of bleeding”. It is because endoscopic intervention is required in both cases. Therefore, patients with active bleeding and “ulcer with a risk of bleeding” should be combined and analyzed.

Response: Following Table 6, we added additional text and new Tables 7 & 8 (pink highlighted text, from lines 202–227). We showed the results of the combined data and added a description of the new results.

6.Conclusion: the conclusion did not answer the research question specifically.

Response: We made corrections by adding the highlighted yellow text from lines 336–341.

Corrections other than those pointed out by reviewers

We corrected Table 1, which was missing anesthesia data and numbers.

We also corrected Table 6, specifically by deleting some items with P values ≤ 0.005.

Round 2

Reviewer 2 Report

Major point

  • Method section: “Fifty-one patients with active bleeding were excluded”
    Why? I think you should first exclude all patients with clinical symptoms findings which indicate active bleeding as the indication of second-look endoscopy was strong for these patients. Only the patients without clinical findings with active bleedings are suitable for this analysis.
  • Results section: Patients with active bleeding was analyzed although the authors mentioned in the method section that they have excluded.
  • Confusing English may be a big issue of this manuscript and it is highly recommended to be intensively revised.

Minor point

  • The abstract does not have the necessary structure required for an original study: it should contain subsections such as aim, method, result and conclusion
  • The definition of “active bleeding” and “ulcer at risk of bleeding” should be addressed in the method section. The definitions were duplicated in the introduction and method sections.
  • Table 1 should be presented in the result section.

Author Response

Major point

  • Method section: “Fifty-one patients with active bleeding were excluded”
    Why? I think you should first exclude all patients with clinical symptoms findings which indicate active bleeding as the indication of second-look endoscopy was strong for these patients. Only the patients without clinical findings with active bleedings are suitable for this analysis.

Response; By removing those with active bleeding rather than excluding them, we aimed to examine risk factors in those who did not bleed. Of the 447 eligible patients, 51 had no clinical evidence of bleeding, but had active bleeding at the time of the second look and were treated for bleeding. Eighty-eight patients did not have active bleeding, but had exposed blood vessels or clots in the ulcer base after resection at the second look, and needed treatment for bleeding when water was applied or clots were removed. It is important to identify risk factors for post-ESD active bleeding or “ulcer with a risk of bleeding,”and therefore patients with either condition were analyzed together.

  • Results section: Patients with active bleeding was analyzed although the authors mentioned in the method section that they have excluded.

Response; We did not simply exclude patients, but analyzed them to identify risk factors, because patients with active bleeding at second look (without clinical findings) should always be treated.

Ulcers at risk of bleeding after resection are those without active bleeding, but with exposed blood vessels or clots at the base of the ulcer after resection at the second look, and bleeding when water is applied or clots are removed, requiring treatment. People with these symptoms are at risk of bleeding if they drink water or eat without being examined. They were analyzed together to identify each risk factor.

  • Confusing English may be a big issue of this manuscript and it is highly recommended to be intensively revised.

Response; Thank you for helpful comment. The English text of our manuscript has again been edited by Editage Co., Ltd. for a fee. English proofreading certificate will be attached.

Minor point

  • The abstract does not have the necessary structure required for an original study: it should contain subsections such as aim, method, result and conclusion

Response; As per your instructions, we have prepared a new abstract with subheadings.

  • The definition of “active bleeding” and “ulcer at risk of bleeding” should be addressed in the method section. The definitions were duplicated in the introduction and method sections.

Response; We eliminated the words in the introduction section.

  • Table 1 should be presented in the result section.

Response; Following your suggestion, Table 1 was moved to result section
as Table 2. Table 2 was changed to Table 1.
